# Intimate Partner Violence against Women during the COVID-19 Lockdown in Spain

**DOI:** 10.3390/ijerph18094698

**Published:** 2021-04-28

**Authors:** Carmen Vives-Cases, Daniel La Parra-Casado, Jesús F. Estévez, Jordi Torrubiano-Domínguez, Belén Sanz-Barbero

**Affiliations:** 1Department of Community Nursing, Preventive Medicine, Public Health and History of Science, University of Alicante, 03690 Alicante, Spain; jordi.torrubiano@ua.es; 2CIBER for Epidemiology and Public Health (CIBERESP), 28029 Madrid, Spain; bsanz@isciii.es; 3Department of Sociology II, University of Alicante, 03690 Alicante, Spain; daniel.laparra@ua.es; 4Interuniversity Institute of Social Development and Peace, University of Alicante, 03690 Alicante, Spain; 5Department of Health Psychology, Miguel Hernández University of Elche, 03202 Elche, Spain; jesus.estevez01@goumh.umh.es; 6National School of Public Health, Institute of Health Carlos III, 28029 Madrid, Spain

**Keywords:** intimate partner violence, SARS-CoV-2, formal help-seeking, Spain, femicide

## Abstract

Aims: To analyze the temporal and geographical distribution of different indicators for the evolution of intimate partner violence against women (IPV) before, during and after the COVID-19 induced lockdown between March and June 2020 in Spain. Methods: Descriptive ecological study based on numbers of 016-calls, policy reports, women killed, and protection orders (PO) issued due to IPV across Spain as a whole and by province (2015–2020). We calculated quarterly rates for each indicator. A cluster analysis was performed using 016-call rates and protection orders by province in the second quarters of 2019 and 2020. ANOVAs were calculated for clustering by province, unemployment rates by province, and the current IPV prevalence. Results: During the second quarter of 2020, the highest 016-call rate was recorded (12.19 per 10,000 women aged 15 or over). Policy report rates (16.62), POs (2.81), and fatalities (0.19 per 1,000,000 women aged 15 or over) decreased in the second quarter of 2020. In the third quarter, 016-calls decreased, and policy reports and POs increased. Four clusters were identified, and significant differences in unemployment rates between clusters were observed (F = 3.05, *p* < 0.05). Conclusions: The COVID-19 lockdown fostered a change in IPV-affected women’s help-seeking behavior. Differences between the volume of contacts made via 016-call and the policy reports generated provide evidence for the existence of barriers to IPV-service access during the lockdown and the period of remote working. More efforts are needed to reorganize services to cope with IPV in non-presential situations. The provinces with the highest 016-call and PO rates were also those with the highest rates of unemployment, a worrying result given the current socioeconomic crisis.

## 1. Introduction

Intimate partner violence against women (IPV) is a complex public health issue encountered across the globe with significant variation in prevalences both between and within countries [1]. In the European context, from the age of 15 upwards, one woman in five has experienced physical and/or sexual IPV, but reported prevalence varies between and within countries [2]. In the 2019 Violence against Women Survey, lifetime physical and/or sexual IPV in Spain was reported by 14% of the women polled, but variation between regions ranged from 20% to 9% [3]. Previous studies have already shown that this variation could be explained by the contextual characteristics of where women live, such as unemployment rates and the levels of gender inequality [4,5,6].

The pandemic caused by SARS-Cov-2 has influenced, and is in turn generating, social inequalities with serious impacts on the population’s health. Pre-pandemic social inequalities—related to poorer-quality housing, unemployment or precarious employment, and restrictions on access to and use of technological resources may be promoting a new social divide. This divide could reproduce and even increase inequalities by gender, socioeconomic status, age, and ethnicity—among other social factors influencing the health and social well-being of the population [7]. IPV constitutes the ultimate expression of gender inequality and also seems to have been aggravated by the current health and social crisis worldwide [8,9,10].

In Spain, the Government Office Against Gender-based Violence noted an increase in the number of telephone calls and WhatsApp messages to the 016-service, which provides consultations and answers requests for information and assistance [11]. Aware of this situation, public institutions have implemented special measures, as other countries have, which include officially declaring services providing IPV assistance as essential. Other measures implemented included transforming tourist accommodation into emergency shelters for women, but this varied according to the specific autonomous community [12]. The differences in the responses produced, along with the already-known variations in case distributions at a national level [9,13], could have influenced the impact of the SARS-CoV-2 lockdown on IPV incidence.

Despite the growing proliferation of publications about both the impact of the COVID-19 crisis and the control measures adopted to manage its impact on IPV, studies using empirical qualitative or quantitative data are still scarce [14]. Most studies that do exist are cross-sectional and focus on evaluating the socioeconomic impact and perception of the seriousness of violent events, using very specific samples (health-service users, affected women and children, and/or women and children from minority groups) with limited follow-up time [15,16,17,18,19,20]. Studies on the impact of the COVID-19 crisis using indicators such as 016-type helpline calls, IPV complaints, and protection measures are still scarce [20,21,22]. These types of indicators provide key information about victims’ ability to access existing resources in this exceptional situation.

The objective of this study was to analyze the temporal and geographical distribution of different indicators associated with IPV. Specifically, we analyzed the evolution of female fatalities, of formal searches or requests for help on the part of the women affected (016-calls, formal complaints), and of the measures implemented (protection orders) before, during, and after the COVID-induced lockdown in Spain from March to June 2020.

## 2. Materials and Methods

We designed a descriptive ecological study based on the numbers of 016-calls, policy reports, women killed, and protection orders issued in relation to IPV across Spain as a whole and by province—including the autonomous cities of Ceuta and Melilla—from January 2015 to September 2020. The 016 telephone-service, provided by Spain’s Government Office against Gender-based Violence, offers information and advice on IPV. In Spain, protection orders are judicial decisions that are enforced via civil and/or criminal injunctions. Among other measures, they include assistance for and social protection of the victim. In this study, we considered the numbers of POs granted, which, in 2020, were 70.9% of the total applied for [23].

Information about the number of 016-calls and the number of women killed was obtained from the statistics portal of the Government Office against Gender-based Violence [24]. Data for official complaints and POs were retrieved from Spain’s General Council of the Judiciary [23]. To calculate the rates, we used the annual population data for women 15 and over from the ongoing municipal census provided by the Spanish Statistical Office [25]; overall unemployment rates, for both men and women by province, for the second quarter of 2020, were obtained from the Spanish Labor Force survey [26]; and finally, IPV prevalence was assessed using microdata from the large-scale 2019 Violence against Women Survey [7].

We calculated, at both national and provincial levels, the quarterly rates of 016-calls, official IPV complaints, POs, and women IPV fatalities. For the first three of these indicators, we calculated quarterly rates per 10,000 women aged 15 and over. For the indicator of women IPV fatalities, we calculated the quarterly rates per 1,000,000 women aged 15 or over. Next, we estimated the inter-annual variation of each indicator—at both national and provincial levels—with the aim of being able to compare each quarterly figure with that of the previous year. The time periods considered to calculate the rates were from the first quarter (Q1) of 2015 up to the third quarter (Q3) of 2020; the second quarter (Q2) of 2020 was that affected by the COVID-19 induced lockdown (14 March to 21 June 2020).

K-Means clustering attempts to identify relatively homogeneous groups of units or entities based on selected characteristics. We carried out a cluster analysis using the K-Means method with the objective of evaluating if the percentage growth in 016-call rates and POs granted between Q2 2019 and Q2 2020 would allow us to identify groups of provinces with distinct behaviors. Female IPV fatalities were removed from this analysis because disaggregating this indicator by province and quarter resulted in a constant with a value of 0. Official complaints were also discarded because they did not include sufficient discriminatory capacity to differentiate clusters. After exploring the data, the cluster analysis was performed without including the autonomous city of Melilla, as it was an atypical case due to the high numbers of 016-calls made and POs granted during the study period.

We evaluated a range of cluster, between 2 and 10, using the different fit criteria available in the R package NbClust [27]. After obtaining a fork of between 3 and 5 possible optimum solutions, we decided on the final number by comparing the reduction in intra-group variability, and, additionally, the increment in the variance explained when adding each new cluster to the solution [28]. Based on the direction and magnitude of the change in rates, the cluster analysis classified the provinces among four groups that showed both greater internal consistency and greater difference from the other groups. Subsequently, we applied an Analysis of Variance (ANOVA) test to be sure that the clusters were significantly different according to the indicators included in the analyses; after establishing that they fulfilled normality requirements via a Shapiro-Wilk test, the homogeneity of variances was then checked using Levene’s test. To establish if there were contextual variables associated with the increase in IPV during the COVID-19 lockdown and the clusters obtained, ANOVAs were calculated considering data clustering by province. The provincial unemployment rates in Q2 2020 and IPV prevalence in the last 12 months were considered as dependent variables. The results concerning the relationship with IPV prevalence may potentially have been influenced by the macro-survey not being fully representative at a provincial level.

We used R-version 4 (R Core Team, Vienna, Austria) [29] and the NbClust-version 3 [27] package for the analyses.

## 3. Results

In the second quarter of 2020, 25,352 016-calls were registered, a total of 34,576 official complaints were made, 5838 POs granted, and there were 4 women IPV fatalities. In this same period, the highest rate of 016-calls was recorded (12.19/10,000) since that of Q3 2016 (12.43/10,000). In the third quarter of 2020, there was a rate of 10.16, similar to that observed in Q3 2015, but higher than Q3 rates during the three previous years (2017–2019). The rates of official complaints (16.62/10,000), POs (2.81/10,000), and fatalities (0.19/1,000,000) in Q2 2020 went down to a level similar to that in 2015, while in Q3 2020, the three indicators all increased (Table 1).

Both in the case of official complaints and in that of POs, the second quarter of 2020 registered the largest decrease for Q2 in comparison with the previous year (−15.2% official complaints and −19.7% for POs). IPV fatalities presented a similar pattern, with a −73.5% decrease compared with Q2 2019. With regard to the percentage of variation between 016-call rates, the growth was almost zero from Q2 of 2015 to Q2 of 2016 (0.1%); it increased very little from 2016 to 2017 (2.4%), with Q2 figures then falling steadily across 2018 (−10.4%) and 2019 (−8.1% in relation to an already lower rate due to the fall in 2018). During Q2 2020, 016-calls registered an increase above 45% as compared to Q2 2019 (Figure 1).

Table 2 shows the inter-annual development of the rates in Q2 2020 as compared to 2019, by province. The 016-calls indicator grew overall, although the high of 155.4% seen in the table is an extreme value corresponding to the autonomous city of Melilla. The next highest value was 118.4%, for the province of Palencia. Despite the overall growth, there were exceptions where negative growth was observed (Soria, Álava, Cáceres and La Rioja), with the Q2 2020 call-rate reaching −27.2% compared to Q2 2019.

Official complaints were quite consistent across the country: they either remained the same or showed a reduction of up to 36%, with just two provinces experiencing increases, Guadalajara (36.1%) and Soria (28.5%). The PO indicator showed a similar trend to that of official complaints, although different provinces that showed positive growth values were identified, up to the extreme value of 184.8%, again found in Melilla and shown in Table 2. In addition, increases in POs were found for the autonomous city of Ceuta (45.3%), five of the eight provinces of Andalusia (up to 30.2%), the Balearic Islands, Las Palmas, and Santa Cruz de Tenerife. During Q2 2020, there were IPV fatalities in four provinces: Jaén in Andalusia, Las Palmas in The Canary Islands, Barcelona, and Girona in Catalonia (Table 2).

Figure 2 shows that the provinces that appear in Cluster 1 registered a decline in both the rate of 016-calls and POs granted in Q2 2020. These provinces were Soria, Cáceres, La Rioja, and Álava, while Teruel and Zamora had zero or almost zero growth in one of the cluster dimensions. The provinces that made up Cluster 2 registered moderate growth in 016-calls (less than 52%) while POs decreased, with Bizkaia presenting the greatest reduction from this cluster. Cluster 3 included provinces that saw an increase in 016-calls during Q2 2020 (more than 60%); POs for this cluster also decreased or, with one exception, stayed constant (Valladolid). Palencia showed the highest growth in 016-calls in this cluster (118%). In terms of POs, Castellón was the province where such orders decreased the most during Q2 2020, while Valladolid remained practically the same compared to Q2 2019. Cluster 4 included the provinces with moderate (11.4%) to high (95.3%) growth in 016-calls and an increase in POs. Ceuta and León stand out, showing the highest growth in both indicators. The areas with the lowest growth in POs granted were Jaén (just below 0), Santa Cruz de Tenerife, Huesca, and Seville. In Huelva, it was also notable that the number of POs granted increased, but not the number of 016-calls. Looking at Figure 2, it is important to note that the four IPV fatalities during this period occurred in provinces that were located in Clusters 3 and 4, where 016-calls and PO rates were the highest.

Statistically significant differences were observed in the total unemployment rate between the clusters (F = 3.05, *p* < 0.05). Specifically, the highest average unemployment rate was observed in Cluster 4, the same for which a simultaneous increase in POs and 016-calls was observed. Statistical differences were also found in the unemployment rate for men (F = 4.66, *p* < 0.01). The lowest average rate was found in Cluster 1 (reduction in 016-calls and POs), the rate increased in Clusters 2 and 3 (rises in 016-calls) and was highest in Cluster 4 (increases in both POs and 016-calls). The unemployment rate for women and IPV prevalences in the different provinces did not show statistically significant differences between the clusters (Table 3).

## 4. Discussion

During the COVID-19 induced lockdown in Spain, there was an increase of more than 45% in 016-calls as compared to the same period in 2019, a volume not seen in any other quarter since 2015. On the contrary, official complaints, POs, and female IPV fatalities decreased, reaching rates not observed since 2015 and early 2016. Subsequently, the rate of 016-calls decreased, and those for official complaints, POs, and killings registered scores similar to those observed in the same period of 2019. In the provinces with the highest general and male unemployment rates, 016-call and PO rates also increased during Q2 2020.

The pandemic caused by the SARS-CoV-2 virus is generating a change in the demand indicators and responses to IPV both in Spain and internationally [22]. This study found an increase in 016-calls during the COVID-19 induced lockdown in Spain between mid-March and June 2020. Some of the factors that could explain this increase in calls were the obligatory confinement with cohabitating abusers and social isolation, along with other factors, such as job instability and the continued presence of minors in the home [30].

The increase in 016-calls during the months of lockdown coexisted, however, with a decrease in IPV complaints, female fatalities, and POs granted [20,21]. Although the courts and the police were considered essential services during the lockdown, access to other sources of support, such as associations or health services, may still have been limited during this period. In the same way, reductions in mobility and social life could have limited women’s access to such services. The decline in fatalities has already been observed in other periods of socio-economic crisis [31], in part, due to the decrease in separations and divorces that usually occur in these periods; these situations can trigger more serious IPV episodes as abusers’ perception of control over women is lower in these situations [32]. More ethnographic information is required about the way in which home confinement has abetted women being controlled by their abusers and how women have dealt with this situation.

Some provinces have been identified in which, not only had POs decreased but also 016-calls, or at least they did not increase. At the other extreme, there are areas in which both 016-calls and protection orders increased. This variation appears to be associated with both general and male unemployment rates, which have also previously been linked to higher IPV prevalence [14,15,16]. The association of male unemployment with greater acceptance of violence at a contextual level, along with greater alcohol consumption, greater individual stress and stress in couples’ relationships are some other factors that could explain this association with IPV [33,34,35,36,37]. Further research is needed to confirm this relationship due to the potential impact that the coming socio-economic crisis could have on IPV prevalence.

The main limitation of this study is that we have carried out an ecological study, which does not allow us to extrapolate the results at an individual level. In addition, the databases used do not provide information that allows other relevant characteristics to be integrated into the analysis, such as individual sociodemographic variables: for both the women affected and their partners, and the characteristics of the environment in which they live. IPV is a complex problem that is influenced by variables of different kinds—individual, contextual, and structural; therefore, its materializations in such an exceptional situation can be very heterogeneous. In this study, however, we have provided a first approximation that needs to be complemented with future studies that can furnish more detailed quantitative data and qualitative information.

## 5. Conclusions

The lockdown and limitation of activities to control the SARS-CoV-2 pandemic seems to have generated a change in IPV-affected women’s formal help-seeking behavior. Our results do not allow us to assess whether actual IPV prevalence has increased; however, it does seem clear that—given the difficulties in accessing the usual sources of formal support—women did seek help through 016-calls. The differences between the number of contacts made through 016-calls and the policy reports generated provide evidence for the existence of barriers to IPV-service access during the lockdown and while working remotely. Greater efforts to reorganize services to deal and cope with IPV in non-presential situations are clearly needed. It is also necessary to study whether there are cases of violence, outside of a pandemic, that are not catered for because the victims are highly restricted in their ability to leave their homes. The trends observed in the indicators analyzed vary between provinces, showing that there are areas in which there has been an increase in both calls and IPV protection orders. These provinces also have the highest rates of general and male unemployment, a worrying result given the current socioeconomic crisis.

## Figures and Tables

**Figure 1 ijerph-18-04698-f001:**
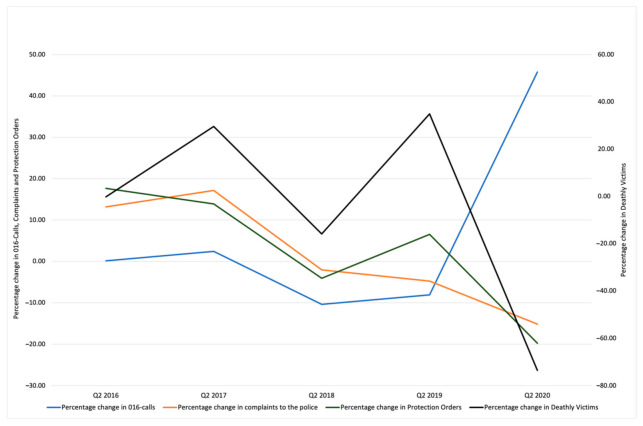
Interannual percentage change in 2nd-trimester rates for the four gender-violence indicators.

**Figure 2 ijerph-18-04698-f002:**
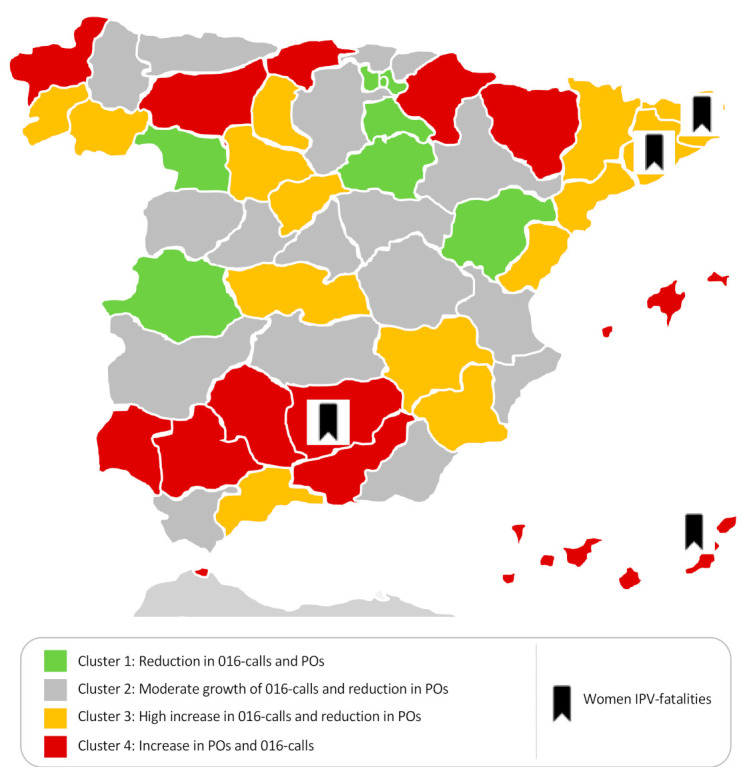
Distribution of Spanish provinces by cluster according to the percentage change in 016-call rates and PO rates in the Second Quarter (2019–2020).

**Table 1 ijerph-18-04698-t001:** Rates of 016-calls, protection orders, and official complaints per 10,000 women aged 15 or over, and mortality data per 1,000,000 for violence against women by their partners, per quarter, 2015–2020.

Year, Quarter	016-Call Rates	Official Complaint Rates	Protection Order Rates	Victim Fatality Rates
2015 Q1	7.78	14.99	2.37	0.45
2015 Q2	9.90	15.85	2.55	0.49
2015 Q3	10.80	16.68	2.76	0.74
2015 Q4	12.11	16.42	2.62	1.19
2016 Q1	10.14	16.75	2.83	0.64
2016 Q2	9.91	17.94	3.01	0.49
2016 Q3	12.43	18.96	3.23	0.49
2016 Q4	9.65	17.23	2.95	0.69
2017 Q1	10.25	19.94	3.15	0.89
2017 Q2	10.15	21.01	3.42	0.64
2017 Q3	9.39	20.95	3.28	0.54
2017 Q4	8.50	19.93	2.97	0.54
2018 Q1	8.64	19.36	3.06	0.34
2018 Q2	9.10	20.57	3.28	0.54
2018 Q3	9.79	21.30	3.41	1.03
2018 Q4	8.38	20.41	3.50	0.39
2019 Q1	7.68	19.50	3.09	0.73
2019 Q2	8.36	19.59	3.50	0.73
2019 Q3	8.76	21.83	3.85	0.53
2019 Q4	8.44	20.43	3.43	0.39
2020 Q1	8.15	17.40	3.10	0.82
2020 Q2	12.19	16.62	2.81	0.19
2020 Q3	10.16	20.60	3.54	0.62

**Table 2 ijerph-18-04698-t002:** Descriptive figures for percentage variations in 016-calls, official complaints, and protection orders for intimate partner violence against women between the second quarter of 2020 and Q2 2019 by Spanish province.

	016-Calls (%)	Official Complaints (%)	Protection Orders (%)
Minimum	−27.19	−36.14	−66.43
Quartile 1	30.92	−21.75	−36.39
Median	46.25	−14.93	−21.50
Mean	47.09	−13.28	−14.33
Quartile 3	66.20	−4.83	0.70
Maximum	155.41	36.09	184.77

**Table 3 ijerph-18-04698-t003:** Means, standard deviations (in parentheses), and ANOVA results for the four variables in relation to provincial clusters for Intimate Partner Violence.

Variables	Cluster 1	Cluster 2	Cluster 3	Cluster 4	F	*p*
Prevalence of IPV	8.01 (3.91)	9.09 (4.95)	10.48 (4.42)	8.62 (5.00)	0.54	0.66
Total unemployment rate Q2 2020	12.15 (4.84)	14.88 (4.10)	13.91 (3.11)	17.84 (5.57)	3.05	<0.05
Male unemployment rate Q2 2020	10.76 (4.22) ^4,^*	12.90 (3.48)	12.39 (2.83) ^4,^*	16.71 (5.04) ^1,3,^*	4.66	<0.01

* Superscript numbers indicate that the mean of that cluster differs significantly from the means of the cluster(s) listed in the superscript, at the 0.05 level in the post hoc test.

## Data Availability

Publicly available datasets were analyzed in this study. This data can be found here: https://www.poderjudicial.es/cgpj/es/Temas/Violencia-domestica-y-de-genero/Actividad-del-Observatorio/ (accessed on 27 April 2021); http://estadisticasviolenciagenero.igualdad.mpr.gob.es/ (accessed on 27 April 2021); https://www.ine.es/dyngs/INEbase/es/operacion.htm?c=Estadistica_C&cid=1254736177012&menu=ultiDatos&idp=1254734710990 (accessed on 27 April 2021); https://www.ine.es/dyngs/INEbase/es/operacion.htm?c=Estadistica_C&cid=1254736176918&menu=ultiDatos&idp=1254735976595 (accessed on 27 April 2021).

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
