# Peer review of "Intimate Partner Violence against Women during the COVID-19 Lockdown in Spain"

_ijerph, 2021, doi:10.3390/ijerph18094698_

Round 1

Reviewer 1 Report

Intimate Partner Violence Against Women During the COVID-19 Lockdown in Spain  

(IJERPH-1167387)

This study aimed to analyze the temporal and geographic distribution of different indicators associated with intimate partner violence. Generally, the study is an important exploration of the noted increase in violence against women during the COVID-19 pandemic. This work is valuable and has several implications. The methodological approach used was also innovative. In light of the importance of this research and the approach used, there remain questions that need to be addressed about the study in general. Therefore, I have decided to recommend major revisions.  Below are some suggestions to improve the article

Abstract

  • The abstract covered the study's purpose, methodology, and results.
  • Perhaps there could be added information pertaining to the conclusion/ implication of the study

Introduction

  • I felt that this section was the weakest part of the manuscript. Although the author(s) provided various justification for conducting the study, there could be more said about the epidemiology of IPV during the pandemic. There are various studies surrounding the increase in violence during the pandemic internationally, and information about the prevalence could be noted.
  • Another issue pertains to the background/contextual information surrounding the issue of unemployment, which the author(s) made a point to include in their assessment of violence and was discussed in the methods section.
  • Some of the paragraphs were seemingly short in the introduction (and throughout) the paper
  • The objective of the study was in general clear

Materials and Methods

  • As noted, the approach used was rather innovative. Although I am not very well-versed in some of the methods used in this study, there are some things I have questions about and require further clarification. I also offer some suggestions
  • I recognize that the author(s) utilized various datasets. I am unclear why the use of data that focuses on population from age 15 and above. If the authors could clarify the use of this age range that would be helpful, given that I assume the focus of the study is on adult victimization. This is generally on a population that is 18 years and older
  • I think it needs to be made clear about the inclusion of unemployment data and how this was used in the analysis
  • Even though the author(s) discussed the different clusters under examination, it might be helpful for the authors to use a chart that highlights the provinces/areas that are within each cluster. Also, the rationale for including certain provinces/areas in a particular cluster. I recognize that a chart of the provinces is provided. However, this was difficult to read. This may have already been highlighted and perhaps may just need clarification
  • I am unclear what is meant by the “significant discrimination”

Results

  • In general, the results were presented well. The author(s) provided key findings of the study clearly

Discussion

  • In general, the research shows there was an increase in calls during Covid-19 compared to previous years, which is highlighted by the author(s).  The results also show a decline in complaints, POs, and fatalities during this period which seems contradictory to the expected findings. Although the authors later offered some rationale for this finding, it raises questions regarding the impact of the pandemic on IPV. Perhaps the author(s) might add other arguments why the hypothesis is supported regarding the increase of IPV within this period
  • There is also an assumption that IPV may have been associated with unemployment. Again, I don’t think the author has outlined this connection early or later in the study.
  • Any other limitations?

Conclusions

  • Perhaps the authors may highlight the implications that this study has, particularly during pandemics.

Author Response

REVIEWER 1

Intimate Partner Violence Against Women During the COVID-19 Lockdown in Spain  

(IJERPH-1167387)

This study aimed to analyze the temporal and geographic distribution of different indicators associated with intimate partner violence. Generally, the study is an important exploration of the noted increase in violence against women during the COVID-19 pandemic. This work is valuable and has several implications. The methodological approach used was also innovative. In light of the importance of this research and the approach used, there remain questions that need to be addressed about the study in general. Therefore, I have decided to recommend major revisions.  Below are some suggestions to improve the article

Abstract

The abstract covered the study's purpose, methodology, and results.

Perhaps there could be added information pertaining to the conclusion/ implication of the study

Thank you for your suggestion. We added this in the current version of the abstract (see lines 28-33).

Introduction

I felt that this section was the weakest part of the manuscript. Although the author(s) provided various justification for conducting the study, there could be more said about the epidemiology of IPV during the pandemic. There are various studies surrounding the increase in violence during the pandemic internationally, and information about the prevalence could be noted.

Thank you for your suggestion. We added more information about this issue in the current version of the introduction (See lines 47-51) in addition to analyses of the existing literature and main gaps made in lines 77-86.

Another issue pertains to the background/contextual information surrounding the issue of unemployment, which the author(s) made a point to include in their assessment of violence and was discussed in the methods section.

We improved the rationale to include the unemployment in our analyses in the current version of the introduction section (See lines 53-66).

Some of the paragraphs were seemingly short in the introduction (and throughout) the paper

Thank to the comments received in this revision, we tried to answer this suggestion providing more detail in the various sections of the manuscript.

The objective of the study was in general clear

Thank you for your comment. We kept it as its original form.

Materials and Methods

As noted, the approach used was rather innovative. Although I am not very well-versed in some of the methods used in this study, there are some things I have questions about and require further clarification. I also offer some suggestions

I recognize that the author(s) utilized various datasets. I am unclear why the use of data that focuses on population from age 15 and above. If the authors could clarify the use of this age range that would be helpful, given that I assume the focus of the study is on adult victimization. This is generally on a population that is 18 years and older

The cut-off age of 15 years is the recommended in the WHO Guidelines for producing statistic on Violence against women (See Guidelines for Producing Statistics on Violence against Women Statistical surveys. United Nations. December 2014. Available at: https://www.un-ilibrary.org/content/books/9789210559874). It is the cut-off age been used by the WHO in its Multicountry study on women’s health and domestic violence against women (See also Garcia-Moreno C, Jansen HA, Ellsberg M, Heise L, Watts CH, WHO Multi-country Study on Women's Health and Domestic Violence against Women Study Team. Prevalence of intimate partner violence: findings from the WHO multi-country study on women's health and domestic violence. Lancet 2006;368(9543):1260-9), and the same used in the questions included in the European Union Agency for Fundamental rights in its Violence against women: an EU-wide survey (Available at: https://fra.europa.eu/sites/default/files/fra_uploads/fra-2014-vaw-survey-main-results-apr14_en.pdf), as well as in the questions included in the Spanish Macro Survey on Gender Violence (Available at: https://violenciagenero.igualdad.gob.es/violenciaEnCifras/macroencuesta2015/pdf/Macroencuesta_2019_estudio_investigacion.pdf) .

I think it needs to be made clear about the inclusion of unemployment data and how this was used in the analysis

We improved the rationale to include the unemployment in our analyses in the current version of the introduction section (See lines 53-66).

Even though the author(s) discussed the different clusters under examination, it might be helpful for the authors to use a chart that highlights the provinces/areas that are within each cluster. Also, the rationale for including certain provinces/areas in a particular cluster. I recognize that a chart of the provinces is provided. However, this was difficult to read. This may have already been highlighted and perhaps may just need clarification.

We change figure 2 and added a map of Spain in order to show clearer the distribution of the provinces between clusters.

I am unclear what is meant by the “significant discrimination”

Results

In general, the results were presented well. The author(s) provided key findings of the study clearly

Discussion

In general, the research shows there was an increase in calls during Covid-19 compared to previous years, which is highlighted by the author(s).  The results also show a decline in complaints, POs, and fatalities during this period which seems contradictory to the expected findings. Although the authors later offered some rationale for this finding, it raises questions regarding the impact of the pandemic on IPV. Perhaps the author(s) might add other arguments why the hypothesis is supported regarding the increase of IPV within this period

As it is now explained in the introduction section, induced lockdown and mandatory confinement with abusers and the consequent social isolation among other factors, such as job instability, are some of the stressors that may have influenced in both, the number of women seeking formal and informal help and the increase of the severity of the events (See lines 47-52). However, in our study, we observed that 016 calls increased but official complaints, POs and female IPV fatalities decreased, reaching rates not observed since 2015 and early 2016. As it is indicated in the current version of the discussion: “The differences between the number of contacts made through 016-calls and the policy reports generated provide evidence for the existence of barriers to IPV-service access dur-ing the lockdown and while working remotely. Greater efforts to organize services to deal and cope with IPV in non-presential situations are needed. It is also necessary to study whether there are cases of violence, outside of a pandemic, which are not catered for be-cause the victims are highly restricted in their ability to leave their homes” (see lines 281-287).

There is also an assumption that IPV may have been associated with unemployment. Again, I don’t think the author has outlined this connection early or later in the study.

As we mentioned before, we improved the rationale to include the unemployment in our analyses in the current version of the introduction section (See lines 53-66).

Any other limitations?

We think the main limitations of the study are included in the current version of the manuscript (see lines 266-275).

Conclusions. Perhaps the authors may highlight the implications that this study has, particularly during pandemics.

Thank you for your suggestion. We added some study implications in the current version of conclusion section: “The lockdown and limitation of activities for the control of the transmission of SARS-CoV-2 pandemic seems to have generated a change in IPV women formal help-seeking behaviour. Our results do not allow us to assess whether IPV prevalence has increased; however, it does seem clear that given the difficulties in accessing the usual sources of formal support, women sought help through 016-calls. The differences between the number of contacts made through 016-calls and the policy reports generated provide evidence for the existence of barriers to IPV-service access dur-ing the lockdown and while working remotely. Greater efforts to organize services to deal and cope with IPV in non-presential situations are needed. It is also necessary to study whether there are cases of violence, outside of a pandemic, which are not catered for be-cause the victims are highly restricted in their ability to leave their homes. The trends observed in the indicators analyzed vary between provinces, showing that there have been areas in which there has been an increase in both calls and IPV protection or-ders. These provinces also have the highest rates of general and male unemployment, a worrying result given the current socioeconomic crisis” (See lines 277-291).

Reviewer 2 Report

Comments for IJERPH-1167387

Thank you for the opportunity to review this paper titled “Intimate Partner Violence Against Women During the COVID-19 Lockdown in Spain.” The current study attempts to fill a gap in the body of literature by using a cluster analysis to assess women’s fatalities, 016-helpline calls, policy reports, and protection orders. The manuscript addresses an interesting topic for scholars who study violence against women, with special attention to its occurrence during a pandemic, and is has the potential to move the body of literature forward. Please see my comments and recommendations below.

  1. My main concern is the “why” for this manuscript. I understand that this study is descriptive in nature and a cluster analysis was conducted as a means to assess the temporal and geographical distribution of different indicators associated with IPV prior to and during the COVID-19 pandemic. That said, the author(s) argument for why this is an important research endeavor was not clearly articulated. Why might the pandemic impact IPV? The authors touch on this in the discussion but a more thorough description in the introduction may strengthen the argument for the paper.
  2. The authors include total unemployment and male unemployment as variables in the current study but do not provide justification for including these variables in the analysis. Why is it important to account for these factors? Additional information in the literature review and/or methods would provide context.
  3. I appreciate that the authors did not overextend the results of the current study. Can any policy implications be made from the results? If would the policy recommendations encompass?
  4. What about avenues for future research?

Thank you for the opportunity to review this manuscript. I hope the above comments and recommendations provide some help when moving forward with this manuscript.

Author Response

REVIEWER 2

Thank you for the opportunity to review this paper titled “Intimate Partner Violence Against Women During the COVID-19 Lockdown in Spain.” The current study attempts to fill a gap in the body of literature by using a cluster analysis to assess women’s fatalities, 016-helpline calls, policy reports, and protection orders. The manuscript addresses an interesting topic for scholars who study violence against women, with special attention to its occurrence during a pandemic, and is has the potential to move the body of literature forward. Please see my comments and recommendations below.

  1. My main concern is the “why” for this manuscript. I understand that this study is descriptive in nature and a cluster analysis was conducted as a means to assess the temporal and geographical distribution of different indicators associated with IPV prior to and during the COVID-19 pandemic. That said, the author(s) argument for why this is an important research endeavor was not clearly articulated. Why might the pandemic impact IPV? The authors touch on this in the discussion but a more thorough description in the introduction may strengthen the argument for the paper.

In the current version of the manuscript, you can read the following: “Violence against women, in any of its manifestations (including intimate partner violence (IPV), sexual exploitation and trafficking of women and sexual harassment, among others), constitutes the ultimate expression of gender inequality, and seems to have been aggravated by the current health and social crisis [2,3]. In relation to intimate partner violence against women (IPV), some of the stressors that may have influenced both the number of women seeking formal and informal help and the increase in the severity of IPV include the covid-induced lockdown and mandatory confinement with abusers and the consequent social isolation, among other factors, such as job instability [4].” (See lines 44-52)

  1. The authors include total unemployment and male unemployment as variables in the current study but do not provide justification for including these variables in the analysis. Why is it important to account for these factors? Additional information in the literature review and/or methods would provide context.

We improved the rationale to include the unemployment in our analyses in the current version of the introduction section (See lines 53-66).

  1. I appreciate that the authors did not overextend the results of the current study. Can any policy implications be made from the results? If would the policy recommendations encompass?

We introduced the following changes in the conclusion section in order to answer this suggestion: “The differences between the number of contacts made through 016-calls and the policy reports generated provide evidence for the existence of barriers to IPV-service access dur-ing the lockdown and while working remotely. Greater efforts to organize services to deal and cope with IPV in non-presential situations are needed. It is also necessary to study whether there are cases of violence, outside of a pandemic, which are not catered for be-cause the victims are highly restricted in their ability to leave their homes” (See lines 281-287).

  1. What about avenues for future research? We made clearer recommendation for future research in lines 253-255; 263-265; and 273-75.

Thank you for the opportunity to review this manuscript. I hope the above comments and recommendations provide some help when moving forward with this manuscript.

Thank you very much for your interesting comments. We are sure that they help us to improve our manuscript.

Round 2

Reviewer 1 Report

This study aimed to analyze the temporal and geographic distribution of different indicators associated with intimate partner violence.  As noted in the previously, this work is valuable and has several implications. The authors addressed many of the previous concerns. In light of this, there are some suggestions/concerns mainly pertaining to the abstract and introduction. I felt that the abstract could be simplified in some areas. Specifically,

Line15: should begin with a Capital after background

Line 16: information included in parenthesis is not necessary

Line 18: beginning with descriptive ecological study seems like an short and incomplete sentence

Line 21: could be stated differently. What is meant by "were calculated considering the provincial membership..

Line 25: what is meant by the "other indicators increased"

Line 31: Beginning with "The provinces with the..........." could be removed

Introduction

Although the authors added some pertinent information, I felt that it was rather disjointed and some information can be rearrange or removed to increase the flow. This  was mainly surrounding the first three paragraphs. 

Materials and Methods

I would recommend that author(s) could begin this sentence differently

Line 97: could be added to the initial paragraph instead of having a separate paragraph. The firs paragraph is rather short.

Line 119: is also a short paragraph and could be added to the previous paragraph

Line 150: this is also a rather short paragraph and could be added to the previous paragraph. 

Results

The map of the country is a nice addition 

General comments

Please ensure that the article is carefully reviewed and edited before submission. 

Author Response

Reviewer 1

Moderate English changes required

Ok. We reviewed the translation

This study aimed to analyze the temporal and geographic distribution of different indicators associated with intimate partner violence.  As noted in the previously, this work is valuable and has several implications. The authors addressed many of the previous concerns. In light of this, there are some suggestions/concerns mainly pertaining to the abstract and introduction. I felt that the abstract could be simplified in some areas. Specifically,

Line15: should begin with a Capital after background

DONE

Line 16: information included in parenthesis is not necessary

DONE

Line 18: beginning with descriptive ecological study seems like an short and incomplete sentence

DONE

Line 21: could be stated differently. What is meant by "were calculated considering the provincial membership..

We rewrite the sentence and know you can read: “ANOVAs were calculated considering data clustering by provinces”

Line 25: what is meant by the "other indicators increased"

DONE

Line 31: Beginning with "The provinces with the..........." could be removed

Thank you for your comment but we should not skip the beginning of the sentence. If we remove it, we would be making an ecological fallacy. Our design is an ecological study, so it is important to refer to our contextual level of analysis that in our case are the Spanish provinces.

Introduction

Although the authors added some pertinent information, I felt that it was rather disjointed and some information can be rearrange or removed to increase the flow. This  was mainly surrounding the first three paragraphs. 

Thank you for your suggestion. We agree with you and we changed this part of the introduction. You can see these changes now in lines 37-54.

Materials and Methods

I would recommend that author(s) could begin this sentence differently

OK, we changed the sentence and now you can read: We designed a descriptive ecological study based on the numbers…

Line 97: could be added to the initial paragraph instead of having a separate paragraph. The firs paragraph is rather short.

DONE

Line 119: is also a short paragraph and could be added to the previous paragraph

DONE

Line 150: this is also a rather short paragraph and could be added to the previous paragraph. 

DONE

Results

The map of the country is a nice addition 

Thank you!

General comments

Please ensure that the article is carefully reviewed and edited before submission. 

Thank you for your suggestion. We did it.

Reviewer 2 Report

Thank you for allowing me the opportunity to review the revised manuscript titled "Intimate partner violence against women during the COVID-19 lockdown in Spain." I appreciate the effort the authors took in addressing my comments and recommendations. I have no further concerns at this time. 

Author Response

Reviewer 2

English language and style are fine/minor spell check required

Ok. We reviewed the translation

Thank you for allowing me the opportunity to review the revised manuscript titled "Intimate partner violence against women during the COVID-19 lockdown in Spain." I appreciate the effort the authors took in addressing my comments and recommendations. I have no further concerns at this time. 

Thank you very much for your suggestions for improving our manuscript.
